# Characteristics of Patients with Laryngomalacia: A Tertiary Referral Center Experience of 106 Cases

**DOI:** 10.3390/diagnostics13203180

**Published:** 2023-10-11

**Authors:** Sergii Bredun, Michal Kotowski, Jakub Mezydlo, Jaroslaw Szydlowski

**Affiliations:** Department of Pediatric Otolaryngology, Poznan University of Medical Sciences, 27/33 Szpitalna Street, 60-572 Poznan, Poland

**Keywords:** laryngomalacia, type of laryngomalacia, supraglottoplasty, stridor, feeding difficulty, comorbidities, synchronous airway lesion

## Abstract

Laryngomalacia (LM) is the most common airway congenital anomaly and the main cause of stridor in infants. Some patients with severe airway symptoms or with feeding difficulties require surgical intervention. Synchronous airway lesions (SALs) may influence the severity and course of the disease. This study aimed to determine the prevalence of various types of LM and SALs and their influence on surgical intervention decisions and feeding difficulties. Moreover, the study focused on the interrelations between SALs and the type of LM or the presence of feeding difficulties. A retrospective analysis of 106 pediatric patients revealed a significant relationship between type 2 LM and the necessity of surgical treatment. We also found a significant effect of LM type 2 on feeding difficulty. Type 1 LM is significantly more characteristic in premature children. Among different comorbidities, SALs are suspected of modification of the course and severity of LM. This study did not find a significant effect of SALs on the incidence of supraglottoplasty or feeding difficulty.

## 1. Introduction

Laryngomalacia (LM) is a collapse of supraglottic structures on inspiration, resulting in a partial airway obstruction. This disorder manifests typically as inspiratory stridor [1] and may be associated with dyspnea, apnea, cyanosis and feeding difficulties with inappropriate weight gain [1,2]. The stridor usually increases in a supine position and during agitation or feeding [3]. LM is the most common airway congenital anomaly and the main cause of stridor in infants [1,3]. Different types of supraglottic collapse in LM are described in the literature [4,5,6,7,8]. A combination of shortened aryepiglottic folds, an omega-shaped epiglottis, an overhanging epiglottis or redundant arytenoid and cuneiform mucosa create various forms of the disorder [1,3].

Awake flexible nasolaryngoscopy (FNL) remains a fundamental diagnostic tool but is insufficient to reveal potential synchronous airway lesions (SAL) distal to the glottis. The regimen of a complete airway examination (FNL and microlaryngotracheobronchoscopy—MLB) for LM is still a matter of debate [4,9,10,11,12,13,14].

LM is a self-limiting disease in the majority of cases, with the symptoms being resolved by the second year of life [1,3]. Nevertheless, some patients with severe or progressive symptoms require surgical intervention. SAL may influence the course of the disease and, thus, surgical decisions. As the incidence of SALs is estimated in 7.5% to 64% [3,12,13,15,16,17] of LM patients, SALs’ role needs further study to establish the criteria for complete airway evaluation.

The primary goal of this study was to determine the characteristics of children diagnosed with laryngomalacia. The specific objectives included the following: (1) to determine the prevalence of various types of LM and SALs based on endoscopic findings (FNL and MLB); (2) to ascertain whether the type of LM influences surgical intervention or feeding difficulties in patients with LM; (3) to examine whether patients with synchronous airway lesions (SALs) have a higher prevalence of feeding difficulties or more often require supraglottoplasty compared with patients without such comorbidity; (4) to assess the relationship between the type of LM and the presence of SAL; (5) to establish whether prematurity modifies any of the above.

## 2. Materials and Methods

This study was designed as a retrospective analysis of all pediatric patients with the diagnosis of laryngomalacia (LM) treated at the Department of Pediatric Otolaryngology, Poznan University of Medical Sciences, Poland. The inclusion criteria were as follows: (1) a hospital admission between January 2015 and June 2022; (2) a diagnosis of LM of any type; (3) a complete endoscopic examination of the airway; (4) and a recorded history of the disease with a prenatal history.

A total of 792 pediatric patients’ medical records were processed. Of these, 123 children with the diagnosis of LM were selected, of which 106 fully met the inclusion criteria. The remaining 17 individuals were excluded from further analysis due to incomplete data, which was a statistical outlier.

LM was diagnosed in children with inspiratory stridor and inspiratory collapse of supraglottic structures observed during a full airway evaluation. The complete endoscopic examination of the airway included a flexible nasolaryngoscopy (FNL) with 2.5–2.8 mm fibroendoscope in an awake patient and a microlaryngotracheobronchoscopy (MLB) using a 0-degree Hopkins rigid endoscope in conjunction with a Miller blade laryngoscope under general anesthesia with spontaneous breathing/ventilation.

Three basic types of LM were distinguished (according to the Monnier classification) as follows: type 1—inward collapse of aryepiglottic folds on inspiration; type 2—curled tubular epiglottis with shortened aryepiglottic folds resulting in circumferential collapse on inspiration; type 3—an overhanging epiglottis that collapses posteriorly, obstructing the laryngeal inlet on inspiration [18]. Some various combined forms (in children presenting more than one type of LM) were also identified.

The medical data were extracted regarding demographics (gender, age, prematurity, birth weight percentile, Apgar score, body weight percentile during the surgery), comorbidities (neurological, cardiac or respiratory, including synchronous airway lesions—SAL) and requirement of surgical intervention (supraglottoplasty).

Feeding difficulty was identified as one of the following symptoms: discoordination of sucking, swallowing and breathing; coughing and choking during feeding (aspiration); worsening of stridor during feeding; failure to thrive.

Statistical analysis was performed using MedCalc software (MedCalc Software Ltd., Version 20.118). The statistical significance level was determined by a *p* value of 0.05 or less. For qualitative indicators (such as the presence or absence of symptoms, surgery, etc.), the chi-square test was used and presented as odds ratios. Also, a regression line was used for the visual interpretation of statistical dependence.

## 3. Results

### 3.1. General

Among the children with LM, there was a clear male predominance, as they constituted 71.1% of participants, and the male-to-female ratio was 2.5:1. Twelve preterm children were identified (11.3%). The mean Apgar scale of the neonates was 9.12 and ranged from one to ten. Birth weights ranged from 530 to 4210 g (mean 3033g). The mean patients’ age during the first hospital admission was 24.75 weeks and ranged from 1 to 82 weeks. The detailed age distribution is presented in Figure 1.

An analysis of the symptoms revealed that stridor was the most common symptom (53.7%). Dyspnea was present in 8.5% of patients; feeding difficulties were observed in 35 patients (33%); body weight deficiency was identified in 4.7% of the children. Four patients were previously tracheostomized due to respiratory collapse.

### 3.2. Surgical Treatment

Thirty-two patients (30%) received specific surgical treatment of LM. At the time of surgery, the mean age was 25.5 weeks (range 2–82 weeks), and the mean body weight was 6940 g. Among surgically treated participants, three were operated on in their first 4 weeks of life, eleven between the ages of 4 weeks and 3 months, five at the age of 3 to 6 months, and thirteen children were operated on at an age of more than 6 months. In total, 24 boys and eight girls (75% and 25%) were operated on. Ten individuals eligible for supraglottoplasty had concomitant diseases, and four of them had SAL. The mean body weight of infants undergoing surgery was 6940 g (range 3100–14,000).

Twelve out of thirty-two surgically treated patients manifested feeding difficulty, consisting of 33.3% of all children with feeding problems.

### 3.3. Type of LM

Type 1 laryngomalacia was revealed in 32 children, type 2 in 51 patients and type 3 in six cases. Combined forms of LM were identified in 17 children (16%): type 1+2 in eight cases, type 1+3 in three cases and type 2+3 in six cases (Figure 2).

Among twelve premature children, eight represented LM type 1, three patients had LM type 2, and one had a combined form of LM (2+3). Type 1 of LM was observed significantly more often in premature children (*p* = 0.0036).

A statistical significance was observed between types 1 and 2 of LM and the conducted supraglottoplasty. Children with LM type 2 were more often treated surgically (*p* = 0.0054), whereas a conventional approach was characteristic for LM type 1 (*p* = 0.0023). Surgery was conducted in 3 out of 32 children with LM type 1 (9.4%) and 22 out of 51 with LM type 2 (43%). Children with LM type 2 consisted of 68.7% of all performed surgeries. 

There was no statistical dependence between LM type 3 and the method of treatment. Similarly, no relation between any of the combined forms of LM and surgical treatment was revealed.

The mean age of the surgically treated children with LM types 1, 2, and 3, including combined forms, was 51.6 weeks, 20.8 weeks, 4 weeks and 35.1 weeks, respectively.

A statistical dependence between the type of LM and feeding difficulties in the children was observed. LM type 2 is significantly more often linked with feeding problems (*p* = 0.0032), whereas children with LM type 1 had no feeding difficulties (*p* = 0.0409). It was found that 47% of all patients with LM type 2 and 18.8% of children with LM type 1 had some feeding difficulties, and 68% of all patients with feeding difficulties were qualified as LM type 2. There was no statistical dependence between other forms of LM and feeding difficulties or between prematurity and feeding difficulties.

### 3.4. Comorbidities

Comorbidities (concomitant diseases) were revealed in 36 children (34%). Almost half of them (44.4%) presented synchronous airway lesions (SALs); this consisted of 15.1% of the study group. Among the comorbid conditions, neurological disorders were identified in four patients, and a congenital cardiac defect was present in seven children.

A posterior laryngeal cleft (LC), including its more extensive variant, called a laryngotracheoesophageal cleft (LTOC), was the most common finding among SALs (61.1%) in children with LM. Three children presented tracheomalacia I or tracheobronchomalacia (TBM) (16.7%); subglottic stenosis (SGS) was identified in one case, vocal fold paralysis (VCP) in one participant and tracheoesophageal fistula (TOF) in one child (5.6%).

Twenty-five percent of children with SAL were operated on. No statistical dependence was revealed between the presence of SAL and the incidence of surgical treatment. Seven out of sixteen children with LM and SAL presented feeding difficulties, but no statistical dependence between SAL and feeding problems was revealed. Among children with SAL, six represented LM type 1; six represented LM type 2; four represented combined forms of LM. There was no statistical dependence between the type of LM and the presence of SAL. Moreover, there was no statistical difference between the mean age of the children at their first admission in children with and in children without SAL. There was no statistical dependence between SAL and prematurity.

The detailed incidence and statistical relations between types of LM, prematurity, SAL and feeding difficulties are presented in Table 1 and Table 2.

## 4. Discussion

Different types of LM have been described in the literature, and several classification systems have been proposed [1,3,4,5,6,7,8,18]. Shah and Wetmore classified LM via the principal site of anatomic collapse: anterior, posterior or postero-lateral [7]. Anterior collapse was caused by the epiglottis; posterior collapse was due to the excess of arytenoid mucosal or cartilaginous bulk. The postero-lateral form of LM was considered to result from the collapse of redundant aryepiglottic folds [7]. This classification system has not gained wide acceptance. In 1999, Olney et al. proposed a classification system that remains one of the most commonly used [4]. Type 1 of this abnormality was described as a prolapse of the mucosa overlying the arytenoid cartilages; type 2 of LM was characterized as foreshortened aryepiglottic folds and type 3 as a posterior displacement of the epiglottis [4]. In 2006, Kay and Goldsmith developed a classification system based on suspected etiology [5]. In their observations, LM type 1 manifested with foreshortened or tight aryepiglottic fold, whereas type 2 was characterized by redundant soft tissue in the supraglottis. LM type 3 was connected by these authors with other etiologies, such as neuromascular disorders [5]. Nevertheless, the former classification proposal has not become as popular and practically useful as those introduced by Olney and Monnier.

The results of our study confirmed the previously reported predominance of males compared to females in LM (2.5:1). Erikson et al. assessed it as 1.9:1 [19], which is consistent with findings of previously published series [1,20]. Simons et al. identified that 17.3% of children with LM were born prematurely [21]. Prematurity was revealed in our study in 11.3% of participants. Although laryngomalacia is not more frequent in preterm infants [15,22], type 1 LM is observed significantly more often in premature children (*p* = 0.004).

LM of a benign course, with mild symptoms presentation, is observed in approximately 40% of children with the anomaly [2]. A similar percentage of patients represents moderate LM and similarly may be managed conservatively. According to the literature, up to 20% of infants with laryngomalacia require surgical management [1,2,4,23]. In our cohort of patients, 30.2% of the children were surgically treated, which is probably a consequence of the highly selected patients in a tertiary referral center. Similar data were published by Glibbery et al. as they operated on 35.4% of their children [24]. On the contrary, only 6.4% of children required surgical intervention in the study of Kusak et al. [25] and 4.2% in the group presented by Wright et al. [20].

Our first hypothesis was that the type of LM influenced the necessity of surgical treatment and feeding difficulties. Ayari et al. regarded feeding difficulties as a marker of the severity of LM but established no strict correlation between a particular type of laryngomalacia and the severity of LM [26]. Our study revealed that children with LM type 2, contrasted with patients with LM type 1, statistically more often require surgical treatment.

According to the results of Kusak et al., nearly 80% of children with LM are at risk of insufficient weight gain [25]. Scott et al. reported feeding difficulties in 86% of their patients [27]. Simons et al. observed symptoms of dysphagia or feeding difficulty in 50.3% of patients [21]. Our study revealed a lower percentage (33%) of children with feeding difficulties. LM type 2 is significantly more often linked with feeding problems than type 1.

The different forms of diagnostic approaches used for children with LM are represented in reports in the literature. Our study protocol included the full endoscopic assessment of the airways in each patient to identify all potential cases of SAL. Glibbery et al. regard microlaryngobronchoscopy as the gold standard for full airway evaluation [24]. Some studies propagated a routine of MLB for all pediatric patients with stridor to reveal the presence of SAL [28,29,30]. Other authors questioned such a position, indicating uncertain clinical significance and low incidence of SAL [13,31,32]. In 2016, the International Pediatric ORL Group published the recommendation to perform an MLB only in patients with severe, progressive or atypical disease [33]. A similar approach was advocated by Olney et al., Masters et al. and Krashin et al. [4,11,13]. Glibbery et al. state that the risk of overlooking SAL in patients with severe, progressive or atypical disease outweighs the potential anesthetic and surgical risks associated with MLB [24]. Nevertheless, there is a scarcity of studies in the literature concerning the evaluation of a selective approach to MLB and the significance of SALs in patients with severe LM [24].

Glibbery et al. identified SALs in 28.2% of their participants, but a limitation of their study was that only 60% of the patients underwent MLB [24]. Rifai et al. reported a SAL incidence of 7.7% but excluded patients with prematurity, cardiac comorbidities and neurological lesions [14]. Krashin et al. provided similar results, performing flexible laryngobronchoscopy solely [13]. Simons et al. observed SAL in 29% of patients [21], and Dickson et al. in 51.7% of cases [12]. Our studies revealed that 15.1% of all participants presented with SAL. In a group of exclusively surgically treated patients, Toynon et al. [16] and Schroeder et al. [15] revealed SAL in 47% and 58%, respectively. These values remain very high when compared to the studies of Glibbery et al. (13%) and our results (12.5%) among patients who underwent supraglottoplasty [24].

The most common SAL in our studies was LC/LTOC (10.37%), followed by TM and/or TBM (2.8%), SGS (0.9%), VCP (0.9%) and TOF (0.9%). Other studies reported TM and/or TBM as the most common SAL, with its incidence ranging from 2% to 48%; this was followed by SGS and VCP [4,9,10,11,12,13,14,34,35]. Simons et al. identified subglottic stenosis in 17.6%, type 1 laryngeal cleft in 4.9%, tracheomalacia in 3.1% and vocal fold paresis or paralysis in 1.8% [21]. 

Landry et al. claimed that SALs have an accumulative effect on airway obstruction [3]. Therefore, our second hypothesis was that patients with SALs are more likely to have feeding difficulties and require surgical treatment. Dickson et al. reported that among infants with mild or moderate disease, those with secondary airway lesions were more likely to require surgical intervention than were infants without secondary airway lesions (27% versus 5.6%; *p* = 0.0002) [12].

Supraglottoplasty was conducted in 25% of children with SAL in our cohort of patients, but no statistical dependence was revealed between the presence of SAL and the incidence of surgical treatment. There was also no statistical dependence between the type of LM and the presence of SAL in our study. There was no statistical dependence between SAL and prematurity.

LC/LTOC was the most common concomitant airway lesion. The high percentage of LC/LTOC among detected SALs in our study may be regarded as controversial. One possibility is that LC is potentially underdiagnosed or overlooked. Parsons et al. and Chien et al. estimated prevalence rates from 6.2% to 7.6% [36,37]. Secondly, the majority of patients with LC/LTOC represent type 1 according to the modified Benjamin–Inglis classification (supraglottic interarytenoid cleft extending down to the level of the vocal cords) [38]. Although this is the most common form of the defect, the diagnostic criteria of LC type 1 are the most vague and subjective. There is no doubt when the LC reaches the level of the vocal folds, but many symptomatic patients only present a significant deepening of the interarytenoid notch. If the cleft involves the cricoid lamina, it represents type 2 or 3a. The term LTOC refers to cases with the involvement of the cervical or interthoracic trachea [18].

Tracheomalacia (TM) was the second most common SAL revealed in our studies. TM is diagnosed during the expiratory phase or during coughing, as more than a 50% bulging of the membranous trachea [18]. Congenital subglottic stenosis is usually a deformity of the cricoid, resulting in a narrowing of the subglottic diameter to less than 3.0 mm or 4.0 mm in a premature or full-term neonate, respectively [18].

Although LC/LTOC was the most common form of SAL in our studies, no statistical dependence between SAL and feeding problems was revealed. Similar observations were made by Simons et al. [21]. On the contrary, Irace et al. reported that a child with a laryngeal cleft and prematurity was statistically significantly more likely to have an abnormal modified barium swallow [39]. Nevertheless, there was no statistical dependence found between prematurity and feeding difficulties in our study. It is possible that the small numbers of patients with SALs, additionally representing a variety of pathologies, did not allow us to detect a significant relationship with the incidence of surgical treatment or the presence of feeding difficulties.

## 5. Conclusions

Type 1 LM is significantly more characteristic for premature children.Children with LM type 2 significantly more often require surgical treatment.There is a significant interrelation between LM type 2 and feeding difficulty.Among different comorbidities, SALs are suspected of modification of the course and severity of LM. This study did not reveal a significant effect of SAL on the incidence of supraglottoplasty or feeding difficulty.

## Figures and Tables

**Figure 1 diagnostics-13-03180-f001:**
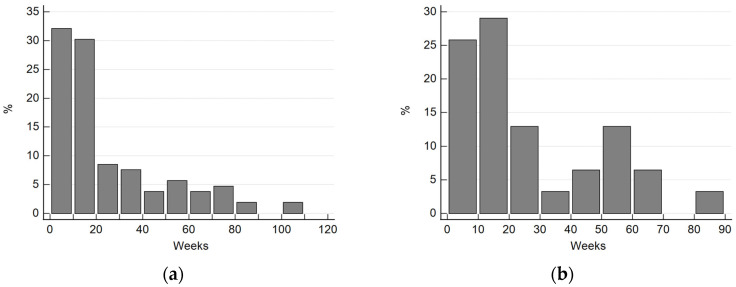
The detailed age distribution in the study group: age of first hospitalization (**a**); age during surgical treatment (**b**).

**Figure 2 diagnostics-13-03180-f002:**
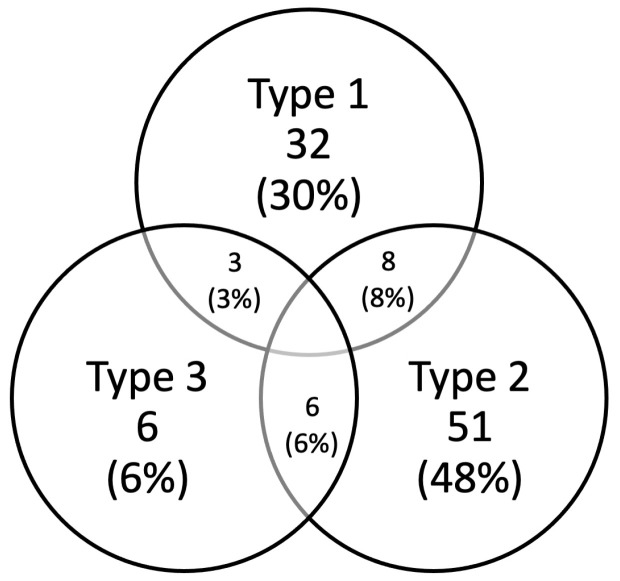
Distribution of different types of laryngomalacia represented in our study.

**Table 1 diagnostics-13-03180-t001:** The incidence of types of LM, prematurity, SAL and feeding difficulties.

	Prematurity	SAL	Feeding Difficulties
All patients	12 (11.3%)	16 (15%)	35 (33%)
Type 1	8 (25%)	6 (18.8%)	6 (18.8%)
Type 2	3 (5.9%)	6 (11.8%)	24 (47.1%)
Type 3	0 (0%)	0 (0%)	1 (16.7%)
Combined types of LM	1 (5.9%)	4 (23.5%)	4 (23.5)
Prematurity		2 (16.7%)	2 (16.7%)
SAL	2 (12.5%)		7 (43.7%)
Feeding difficulties	2 (5.7%)	7 (20%)	

**Table 2 diagnostics-13-03180-t002:** The statistical relations between incidence of types of LM, prematurity, SAL and feeding difficulties.

	Prematurity	SAL	Feeding Difficulties	Surgical Treatment
LM type 1	*p* = 0.0036	*p* = 0.4914	*p* = 0.0409	*p* = 0.0023
LM type 2	*p* = 0.0903	*p* = 0.3588	*p* = 0.0032	*p* = 0.0054
LM type 3	*p* = 0.3698	*p* = 0.2899	*p* = 0.3828	*p* = 0.8635
Combined forms of LM	*p* = 0.4421	*p* = 0.2913	*p* = 0.3662	*p* = 0.9396
Prematurity	-	*p* = 0.8723	*p* = 0.2030	*p* = 0.6790
SAL	*p* = 0.8723	-	*p* = 0.3242	*p* = 0.6253
Feeding difficulties	*p* = 0.2030	*p* = 0.3242	-	*p* = 0.5208

## Data Availability

Data available from the corresponding author upon request.

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
