# Peer review of "Characteristics of Patients with Laryngomalacia: A Tertiary Referral Center Experience of 106 Cases"

_diagnostics, 2023, doi:10.3390/diagnostics13203180_

Round 1

Reviewer 1 Report

This manuscript of a retrospective study on laryngomalacia and synchronous lesion in pediatric patients is well written regarding format and scientific aspects.

The quality of English in the manuscript is good; minor edition is required.

Author Response

Response to Reviewer #1

Reviewer’s comment:. This manuscript of a retrospective study on laryngomalacia and synchronous lesion in pediatric patients is well written regarding format and scientific aspects.

Reply: I would like to thank the Reviewer for the revision. I appreciate your assessment of the study.

Reviewer’s comment: The quality of English in the manuscript is good; minor edition is required.

Reply and action: Thank you for this comment. The manuscript has been checked and some minor corrections have been made.

Reviewer 2 Report

Thanks for the opportunity to review this retrospective review on Laryngomalacia and its outcome from  a Paediatric ENT unit in Poland. 

A few questions for clarification

1. What was the clinical severity of the problem in this cohort of babies?

2. A table on detailing surgical details- indication, procedure

3. What was the outcome of surgery and non-operative management in this cohort?

Many of the references are very old. 

Author Response

Response to Reviewer #2

Reviewer’s comment: Thanks for the opportunity to review this retrospective review on Laryngomalacia and its outcome from  a Paediatric ENT unit in Poland. 

Reply: I would like to thank the Reviewer for the revision and valuable comments.

Reviewer’s comment: A few questions for clarification

  1. What was the clinical severity of the problem in this cohort of babies?
  2. A table on detailing surgical details- indication, procedure
  3. What was the outcome of surgery and non-operative management in this cohort?

Many of the references are very old. 

Reply and action: I agree that some of the references are old but there are two main reasons. The first is that the study included some historical background of different classification systems of LM. The second is that there was a limited number of publications corresponding precisely to the aim of our study.

Responses to your questions:

Ad.1

The clinical severity of LM was variable in the study group as the study covered all pediatric patients with the diagnosis of laryngomalacia (LM) treated at the Department of Pediatric Otolaryngology, Poznan University of Medical Sciences, Poland between 01.2015 and 06.2022 meeting the inclusion criteria. The children with severe symptoms of respiratory distress or feeding difficulties were eligible for surgical intervention.

Ad.2

With all due respect, I am not convinced about the additional table presenting purely surgical results. Discussing the details of different types of supraglottoplasties might be intricate for many readers. On top of that, the assessment of surgical treatment results was not the aim of this study. But undoubtedly it could be the subject of another manuscript. The extent of the surgery in laryngomalacia depends strictly on the type of LM.

According to the literature and clinical experience, there are two main indications for surgery in children with laryngomalacia: inadequate body mass gain due to feeding difficulties and respiratory distress (dyspnoea or severe stridor). All the patients from this study were qualified to supraglottoplastic procedures according to these criteria.

Ad. 3

Similarly to Ad.2, the assessment of surgical treatment results was not the aim of this study.

Nevertheless, the results were as follows: none of the patients required tracheostomy; the one stage surgery was sufficient for the majority of patients. Regarding the natural course of the disease, the conventional treatment

The lack of improvement or deterioration of symptoms in conventionally treated patients with LM was the indication for supraglottoplasty.  It was also one of the reasons of wide age range in the study group. Some of the children who were surgically treated in our department had been previously unsuccessfully surgically treated in different pediatric Centers.

I hope you will find my explanations satisfactory.